# Enhancement and Communication of Ancient Human Remains through VR: The Case Study of Sexual Dimorphism in the Human Skull

Roberta Manzollino [1], Saverio Giulio Malatesta [1], Danilo Avola [2], Luigi Cinque [2], Antonietta Del Bove [3,4], Laura Leopardi [1] and Marco Raoul Marini [2,*]

1    Interdepartmental Research Center DigiLab, Sapienza University of Rome, 11085 Rome, Italy; roberta.manzollino@uniroma1.it (R.M.); saveriogiulio.malatesta@uniroma1.it (S.G.M.)
2    Department of Computer Science, Sapienza University, 00198 Rome, Italy
3    Department of History and History of Arts, University Rovira I Virgili, Avinguda de Catalunya 35, 43002 Tarragona, Spain
4    Catalan Institute of Human Paleoecology and Social Evolution (IPHES-CERCA), Edifici W3, Campus Sescelades URV, Zona Educational 4, 43007 Tarragona, Spain
*    Correspondence: marini@di.uniroma1.it or marcoraoul.marini@uniroma1.it

**Abstract:** Over the last years, the exponential progress of technology introduced a broader population of researchers and developers to the use of Virtual Reality (VR) devices in numerous contexts, e.g., gaming, simulations, and culture dissemination. Recently, cultural heritage has also been supported by motivational experiences and other improvements designed explicitly for specific users (visitors, researchers, and domain experts). In this context, we propose a protocol within a digital environment, using innovative, non-invasive, and non-destructive methods for the technological enhancement, education, and dissemination of ancient human remains. The presented case study is focused on sexual dimorphism in the human skull; several 3D models are digitally generated from female and male skull references exploiting an algorithmic approach with statistical analysis, e.g., Principal Component Analysis (PCA); then, the models are made available in a virtual environment with a Head Mounted Display (HMD) and can also be interacted with via a touchless approach (hands-free). Tests conducted with segmented populations provided promising results.

**Keywords:** virtual anthropology; sexual dimorphism; BioAnthropology; VR; user experience





## 1. Introduction

Biological anthropology is the discipline that studies archaeological human remains to obtain information about the lifestyle of ancient populations. Skeletons represent the most direct evidence of past populations [1]. The anthropologist is a highly specialized person who routinely handles human remains, such as skeletons or mummies, to perform the necessary analyses. In particular, bone manipulation is sometimes very important; the rotation and the visualization from all perspectives could help better understand the morphology of a skull, for example. One of the tasks of the anthropologist is to determine the biological sex of the individual. The skeletal regions with the most significant degree of sexual dimorphism, having the most pronounced morphological variation between the sexes, are the skull and the pelvis [2]. While this might seem obvious to professionals in medicine and human anatomy, it is not to an audience of children, teenagers, or ordinary museum visitors. Moreover, understanding this kind of topic could be complex in a specific context; museums that display human remains offer visitors a non-interactive experience, usually limited to simple observation. The specimens are placed behind glass with no opportunity to interact with, touch, or manipulate them for conservation purposes, as with any museum exhibit, as well as ethical concerns [3]. Ancient human remains have high symbolic, emotional, cultural, and religious value, and therefore, they are considered

"culturally sensitive artifacts", but they are also the subject of great interest among the non-specialist public, experts in the field, and the scientific community. However, the status of what constitutes "human remains" is not yet fully defined. The scientific debate about the exhibition of human remains and the related ethical issues are thus increasing, and some international regulations (e.g., ICOM) are beginning to govern their care and exhibition in museum collections; it is necessary to respect both the descendant communities to whom the remains belong and the personal sensibilities of the visitor [4]. In museums, that is reflected in the interest in finding suitable solutions for displaying these sensitive specimens and understanding the visitor's feedback when approaching human remains, including multimedia. The development of computerized techniques and their adoption in this field allowed us to visualize and analyze human remains in a virtual environment, so-called Virtual Anthropology (VA) [5].

The VA experience uses digitization, manipulation, restoration, and conservation methods for studying, preserving, and communicating cultural and biological artifacts [6]. The digitization of specimens allows the study of human collections in total conservation security, leading to significant advantages in educational, dissemination, and enhancement aspects, particularly in museums [7]. In the same way, the exponential progress of technology introduced a broader population of researchers and developers to the use of Virtual Reality (VR) devices in numerous contexts, e.g., gaming, simulations, sharing of knowledge, and cultural traditions. In this context, human–computer interaction techniques have been widely investigated through the years [8–10], particularly for dissemination and teaching purposes [11]. In fact, since the dawn of modern computer science interaction technologies, e.g., wearable devices, camera-based body trackers, or gaze recognizers, cultural heritage, and museums have taken advantage of their use [12]. VR and Augmented Reality (AR) technologies are widely exploited in related application areas and experiences [13], e.g., improving museum visiting or allowing researchers to deal with digital replicas of archeological finds and anthropological specimens and those advanced high-level interaction systems are significantly employed in recent research [14–16]. Moreover, they allow for overcoming barriers in accessibility issues [17], improving features for simplifying physical and logical tasks. VR also allows viewing and studying inorganic cultural artifacts (i.e., coins, statues, vases) [18,19] or human remains (bones, mummies) [20] in more detail and depth than in the classic museum display. In addition, with VR devices, it is also possible to actively interact with them through manipulation, rotation, movement, and the ability to look inside them in an immersive way. Immersive VR devices, such as Head Mounted Displays (HMD), are nowadays equipped with motion sensors that allow users to naturally interact with the virtual environment by moving around the room.

In this paper, we developed a research protocol and an experience that includes some principles of serious gaming using VA, VR, and engagement methodologies to explain some principles of biological anthropology, such as the sexual dimorphism of the human skull. In detail, the combined approach of multiple techniques allowed us to show how this technology improves the museum experience and innovatively helps the understanding of osteological concepts. In this vein, the proposed work provides a further step in the usability and user experience field by combining accurate storytelling with advanced interaction technologies. Indeed, one of the challenges of the museum is to make its collections immediately understandable to a non-specialist audience. This challenge becomes even more daunting when it comes to human remains, ancient and often of archaeological origin.

## 2. Related Work

VA is an interdisciplinary field of biological anthropology: it brings together experts from different domains such as anthropology, biology, medicine, mathematics, statistics, computer science, and engineering. VA exploits digital technologies to study human morphology, mainly the human skeleton [5]. Indeed, it is possible to create virtual replicas of human skeletons or bones, which can be used to analyze bone shape, structure, and characteristics [21]. These models can be created using laser scanning, photogrammetry,

or computerized tomography (CT-scan) techniques, which can then be processed using specialized software to create the 3D digital model. It is used in various applications, including scientific research and dissemination. Consequently, VA allows the digital study of ancient human remains [6], and it operates in six areas (Table 1) [22].

**Table 1.** The six areas of VA and their description as explained by Gerhard W. Weber [22].

| Operational Areas of VA | Description |
| --- | --- |
| Digitize | Mapping the physical word |
| Expose | Looking inside |
| Compare | Using numbers |
| Reconstruct | Dealing with missing data |
| Materialize | Back to real world |
| Share | Collaboration at the speed of the internet |

Hence, as opposed to the traditional approach in which an anthropological study is carried out on the original skeletal elements, it is possible to perform analysis on a computerized 3D model of the original specimen through VA techniques.

There are many advantages to having accurate 3D models available for research, conservation, enhancement, educational, and dissemination purposes.

The entire structure is accessible, and many analyses (e.g., volumetric, form, measurements, visualization, etc.) can be performed and replicated [22]. One of these analyses is Geometric Morphometrics (GM).

GM is a methodological approach for acquiring, processing, and analyzing data for studying and comparing biological shapes transformed into numbers [23]. The fundamental concept is based on geometrical anatomical 'homology' between two related forms based on ontogenetic or phylogenetic criteria. GM allows us to carry out multivariate statistical analysis and quantitatively analyze the shapes in their complexity as a whole rather than a set of individual linear measures or indices [24–26].

The use of VA has several advantages from a conservation perspective; the digital replica could be virtually manipulated, thus preserving the original specimen and safeguarding it from the risk of mechanical damage and chemical and physical alterations. Moreover, it makes it possible to perform virtual restoration to repair damaged or missing regions [27,28].

VA also plays a crucial role in communication and educational aspects. The 3D models produced, which can be accurate replicas of the original artifacts or purpose-made virtual objects, can be used in museum contexts. For example, they can be uploaded to a virtual environment and become usable on various devices to facilitate learning while entertaining the user.

In Italy, many categories of museums, such as archaeological, anthropological, human anatomy, and pathological anatomy museums, exhibit essential collections of human remains. The cultural heritage represented by human remains in Italian museums is abundant but sometimes dispersed and poorly enhanced. [4].

In recent years, museology has adopted interactive and virtual techniques for exhibition, communication, and education, succeeding in reaching a broader and more diverse audience and trying to involve younger age groups more. In addition, multimedia content is key in enhancing and disseminating scientific information [12]. The use of advanced technologies, gamification, and the support of innovative storytelling methodologies can contribute to the appreciation and understanding of human remains in museum collections and to the discovery of life history of individuals who lived in the past. Compared to observation alone, this approach offers visitors an alternative way to appreciate and learn more about ancient human remains from an osteological and archaeological perspective. It also offers the opportunity to deepen aspects invisible to the naked eye, to learn about and

graphically visualize the multidisciplinary studies behind their display, respecting both the human remains' dignity and the public's sensibilities [5].

Although these 3D technologies have been widely used for decades for the documentation, preservation, and communication of cultural heritage due to their accuracy and precision [19], few technologies are applied to the touch and appreciation of human remains in Italy. The most notable examples are the South Tyrol Museum of Archaeology's "Ötzi the Iceman", the temporary exhibition "Invisible Archaeology", and the permanent hall "In Search of Life," both housed in the Egyptian Museum in Turin.

At the South Tyrol Museum of Archaeology (https://www.iceman.it/ (accessed on 1 April 2023)), visitors can learn about the original findings of the Ötzi mummy, his history, and his previous scientific research, through virtual and interactive stations. Virtually, there is a choice of what to look at in the 3D model of the mummy: the outer surface, the skeletal system, or performing a virtual autopsy. In these examples, technologies have made it possible to make visible what is invisible.

In the temporary exhibition "Invisible Archaeology", using 3D technologies, it was possible to perform virtual unwrapping and reveal what is hidden inside the coffin of the mummies Kha and Merit: not only bones and organs but also jewelry and funerary ornaments. A virtual tour of the exhibition was also available online, making it possible to explore the exhibition rooms, browsing all the elements, from videos to individual exhibits, from any device.

"In Search of Life" is the new permanent exhibition room dedicated to life in ancient Egypt by studying human remains, focusing on six mummies representing the fundamental stages of life. In this case, the exhibit display is not interactive but is observed passively through monitors in which it is possible to see both the virtual unwrapping of the mummies with related insights (https://museoegizio.it/ (accessed on 1 April 2023)).

These exhibits are very popular with audiences as they offer visitors an alternative means to discover the lives behind museum specimens of human remains.

However, no "a posteriori evaluation" of these applications' impact on museum visitors—neither from the point of view of popularity nor usability—has been conducted or published.

### 3. The Case Study of Sexual Dimorphism in Digital Environments

As previously mentioned, VA and VR could enrich the visitor experience toward the display of human remains. With the use of these techniques, the passive observation of the specimen can be supplemented with the interactive participation of the visitor who can touch to manipulate digital replicas of human remains in a virtual and immersive environment. This approach leads to a significant improvement in didactic and educational aspects [29]. The contents of a museum are generally offered to visitors through installations correlated by text explanations (e.g., through information panels) or sound (e.g., through audio guides made available at the entrance). At the same time, the itineraries of the exhibitions are left free to the visitor, possibly suggested by indicators. Alternatively, visitors can be accompanied by guides offering them a more in-depth cultural experience, but this may involve restricting the visit to specific content, not satisfying to most visitors. Indeed, it also makes a cultural visit a playful and entertaining experience thanks to gamified content in exhibits being a stimulus and facilitating the active interaction with the museum by expanding the audience of users, with particular reference to younger visitors. This choice is validated by previous studies confirming that gamification is increasingly utilized to promote cultural heritage [29,30].

In this paper, the case study focuses on sexual dimorphism in the human skull and the morphological differences between the female and male cranium. The human cranium exhibits some anatomical features different between sexes [31]. For example, the mastoid processes are larger in males, the glabellar region is smoothed in females, and the brow ridge is more developed in male individuals [32,33]. These are some different features used to determine male or female identification (Figure 1). Traditionally, sex identification in

human skulls is performed by measuring the presence or absence of discrete morphological traits or identifying their expression rate.

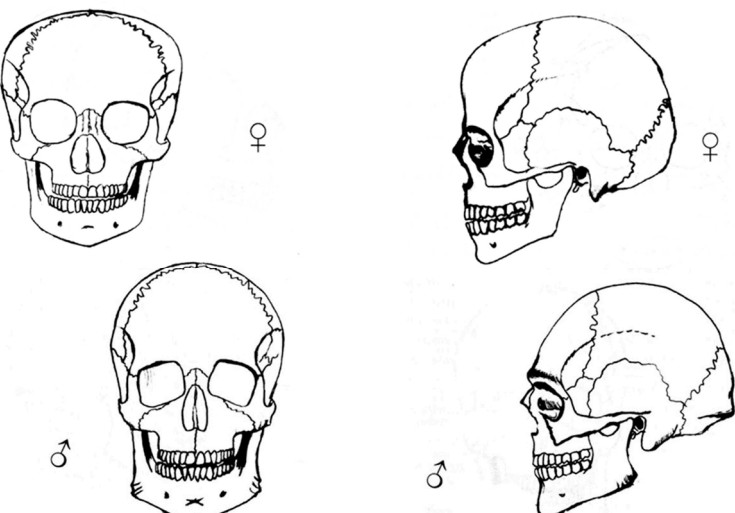

**Figure 1.** A female skull (**top**) and a male skull (**bottom**), in frontal and lateral view, compared (image modified from [2]).

These differences are often difficult to interpret, even for specialists. Human skulls do not always exhibit features that can be readily classified using traditional methods. Furthermore, dimorphism is less pronounced in some individuals and populations than in others, where it is easily recognizable [2]. Thus, if recognizing these traits is challenging for biological anthropologists, it is even more complicated for non-specialist audiences.

## 4. AnthroVR: A System for Exploring Anthropological Remains with VR-Based Technology

Based on the background knowledge discussed in Section 3, we developed a system for improving the dissemination of anthropological content to non-specialized users. There are very limited boundaries about the target audience: anyone should enjoy the experience, from children to adults, from non-experts to anthropologists, and from VR-accustomed users to unprepared users. However, people with disabilities (e.g., blind, deaf, severe cognitive deficits, or motor problems affected) will be included in future study improvements.

In this context, the system is developed with two different environments: the skulls 3D editor, namely MoldeR, and the virtual experience one.

### 4.1. MoldeR

We present the tool MoldeR developed in the R language [34], version 3.5.3. MoldeR has considerable potential in constructing casts molded according to specific information derived by categorical groups. This tool could be used to define 3D models starting from a template, e.g., the surface of a human skull. We applied VA to an archaeological collection to create 3D models corresponding to female and male cranial morphology but with the associated sexually dimorphic features magnified. In this way, the morphological differences between the skulls of the two sexes are immediately evident. Specifically, we developed a protocol to extract the morphological component linked to a categorical variable (biological sex in this case study) from landmark configuration acquired on a collection of adult skulls of known sex.

The study sample consisted of 163 skulls of adult individuals, 74 females and 89 males, belonging to four digital osteological repositories: the Lynn Copes Digital Collection (Black Americans) from the Anthropology department at the National Museum of Natural History, Washington, DC; the Museum of Anthropology "G. Sergi" from Sapienza University of Rome; the Oloriz Collection in Spain; and the Anthropological Museum of Florence

(University of Florence, Firenze, Italy). The 3D models were obtained via photogrammetry and CT scan [33].

The sample has been divided into two groups according to sex variables. On each skull, we recorded the 3D coordinates of 50 anatomical points (landmarks) and 200 geometrical points (semi-landmarks) using Amira Software by Termofischer Scientific. All points are distributed around the entire cranium to acquire better anatomical details: the facial complex, the neurocranium, and the cranial base.

MoldeR is code developed in the statistical environment (R). The first step is a Generalized Procrustes Analysis (GPA) to translate, rotate and scale the landmark configurations on the mean shape removing differences in spatial positioning, orientation, and size. Subsequently, the shape variables (the landmark configurations after GPA) are analyzed using multivariate statistical analysis, Principal Component Analysis (PCA). A sample of the landmarks and their configuration is reported in Figure 2. The result of PCA is shown in the related work of Del Bove et al., 2019 [32]. The average shape (target) associated with each level of the categorical grouping is calculated from the breakdown of the PC scores. The researcher can define a reference model to be warped into the constructed 3D model using the Thin Plate Spline (TPS) algorithm [35]. The base concept is mapping homologous points between different shapes using a distortion grid. TPS algorithm enables a visual representation of how one shape can be morphed into another through grid distortion. A function (*spline*) is then computed to interpolate a set of points in an interval so that the function is continuous.

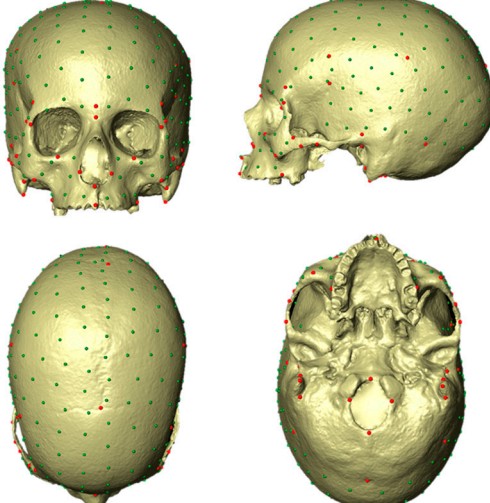

**Figure 2.** Landmark (in red) and semi-landmark (in green) configuration. We recorded the 3D coordinates of 50 landmarks and 200 semi-landmarks on each skull to acquire the entire morphology.

TPS is a spline-based interpolation algorithm used here to create warped surfaces. In VA, TPS is widely used to interpolate missing data using homologous landmarks as a reference. In this case study, we apply TPS to create representative female and male cranial morphology meshes [25].

We calculated two 3D models associated with the female and male cranial shapes. MoldeR allowed for the reshaping of skulls starting from a chosen reference model into a targeted one obtained by calculating the mean shape of a given group (e.g., female and male sub-groups). In this application, we applied the PCA to select only PC scores related to sexual dimorphism. Selected PC scores have been used to build mean representative models of mean female and male cranial morphology.

Magnification was performed in two ways. First, we built two models showing the actual differences in shape related to sex. Second, we increased the magnification of the shape variation by a factor of 2. We applied a posterior cross-validation test to test the statistical significance of the separation between female and male morphology. The

accuracy for distinguishing between females and males using this methodology from PCA is equal to 78% (males = 80%, females = 75%), according to the literature [33]. The shape variations depicted in Figure 3 reported only the different anatomical traits between females and males obtained via PCA.

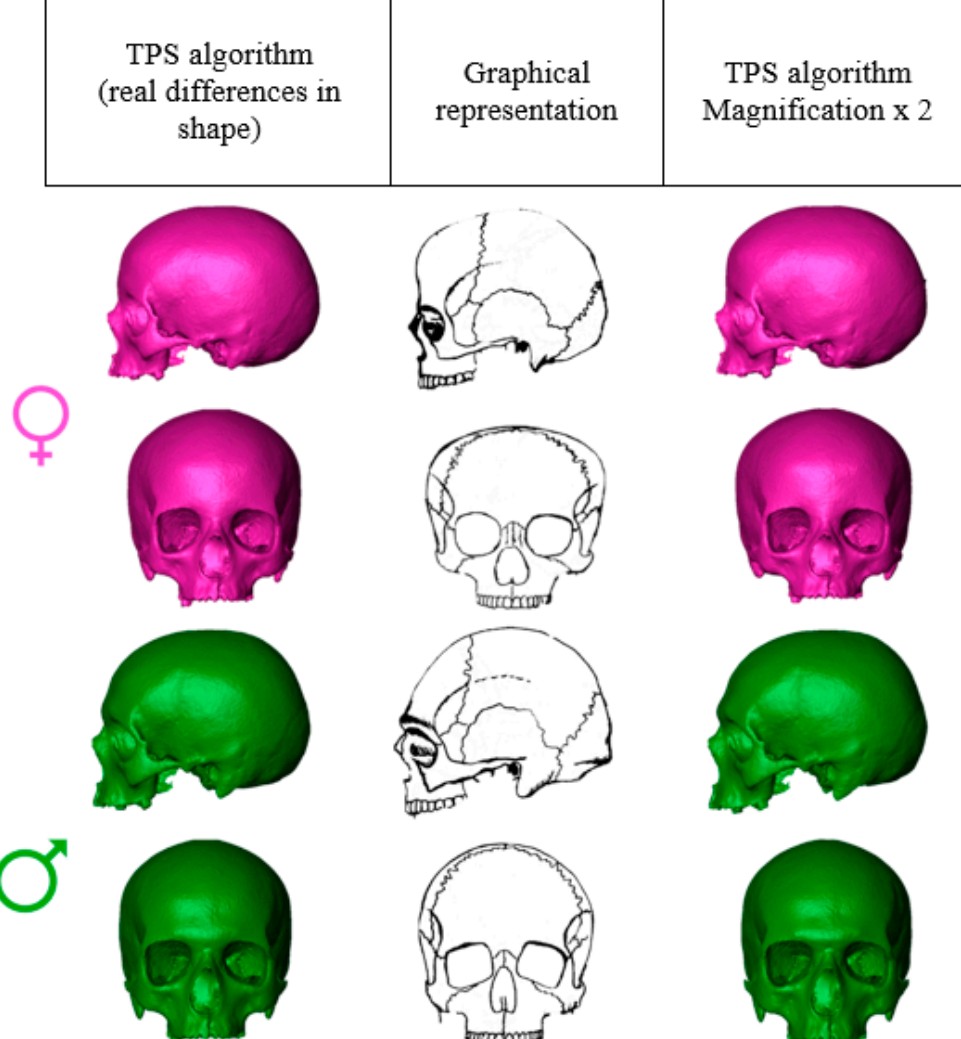

| TPS algorithm (real differences in shape) | Graphical representation | TPS algorithm Magnification x 2 |
| --- | --- | --- |

**Figure 3.** 3D models associated with the female (purple) and male (green) cranial shape with a factor of magnification of 1 and 2 (on the left and the right, respectively). As usual in GM, size is not considered here.

## 4.2. The Simulator Architecture

The system is developed in two phases: the content and the interaction setup. In the first step, the virtual environment was populated with the scene elements, first the static and then the elements users could interact with. The full context was previously designed on paper with a storyboard to propose an immersive experience with a serious game approach. Then, in the second phase, the logic of the serious game is deployed, following the previously hypnotized storytelling.

The involved devices are the Oculus Rift CV1, the LeapMotion, and a high-end desktop computer with the following specifications: CPU Intel i7 5930 k, RAM 32 GB DDR4 3200 MHz, Nvidia GeForce RTX 2080 ti, Motherboard Asus Rampage V Extreme, and a storage SSD Samsung 860 Pro 1 TB.

The architecture of the system is shown in Figure 4.

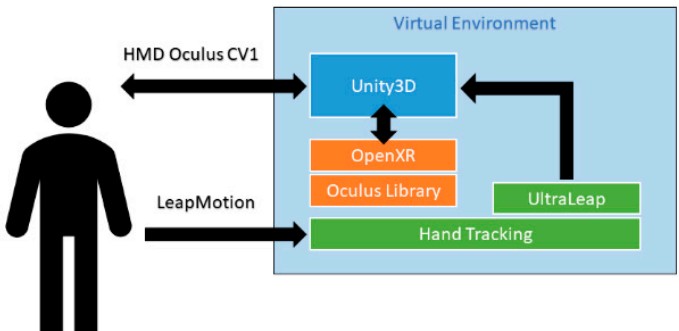

**Figure 4.** The architecture of the proposed system. The HMD directly provides and receives information to Unity3D, which exploits OpenXR and Oculus libraries for communicating with the device. At the same time, the frontally mounted LeapMotion captures the input from the hands with the IR cameras and, through the UltraLeap library, infers the joints' positions, which are then provided to Unity3D.

As shown, two flows can be identified: the input/output of the HMD and the hands-tracking thread. The first involves the physical HMD device directly connected to the desktop computer with USB 3.0 and High-Definition Multimedia Interface (HDMI) cables. The central core of the system is a Unity3D environment deployed for a Windows-based system. The software also requires the usage of two Infrared (IR) tracking sensors to allow accurate head position tracking of the user; this explains why such a specific motherboard, provided by ten back-panel USB 3.0 ports, is involved in the experimental setup. The environment exploits two libraries for correctly communicating with the HMD: OpenXR and Oculus Integration. Both are specifically designed for VR applications in Unity3D. Still, the second add specific functions to the first to improve performance and compatibility with Oculus-branded (owned by Meta) products.

The second flow involves the LeapMotion, a small device with two IR cameras to track hands. It can be placed in front of an HMD with a mounting plug for recognizing elements in ego-view while the user wears the helmet. The software of this device has consistently evolved since the first stable version (around 2015); the latest release, Gemini V5, is designed explicitly for VR purposes, improving the tracking quality and the algorithm's robustness. The movements of the 3D reconstructed hands with joints and bones are smooth also in hostile environments, e.g., low/high artificial illumination conditions, other IR signals, or close to windows during a sunlight day.

The software exploits the previously mentioned libraries and two custom functions: Object Grabbing and Environment Resizing. The first algorithm consists in detecting the distance between the tips of the thumb and the tip of each other finger, according to a specific equation[1].

Gravity is ignored for improving usability and simplifying the interactions, and each game object is kinematic (no forces are applied).

Environment Resizing is a function that changes the scale of the virtual environment according to the user's height. It is essential to consider usability with very heterogeneous users, particularly in the case of a tall adult and a child. In this context, the proposed solution allows each participant to enjoy the experience without running a calibration phase. Starting from a proportion 1:1 when a user is 1.70 m tall, the algorithm detects the distance from the ground of the HMD when the application is executed and proportionally scales the environments and its GameObjects, except the skulls, which are translated on the correct $y$ value to be easily grabbed. This detail is relevant for keeping the models as realistic and authentic as possible for the user.

*4.3. Experimental Environment*

We developed a task for a serious game where the 3D skulls were used to show human sexual dimorphism in an immersive and interactive manner. The experience is divided into

two phases and is set in a futuristic environment. A robot named RobSaRa, is the guide and the narrative voice. The experience requires the supervision of an operator to support the user both in wearing the HMD and verifying the correct execution of the run.

The first phase is purely introductory and provides the necessary directions to move around the virtual space and interact with objects in the scene. The guide's voice prompts the user to look at his/her hands and keep them in the scene, which means in front of his/her face. Then it explains that to pick up an object, it is necessary to close the hand, while it must be fully opened to let it go. After this stage, the user enters the main scene. He/she is invited to explore the virtual environment visually, looking first to the right, then to the left, and finally behind him/her. In the front view is a desk with a monitor with the Sapienza logo, a notebook, a pen, a flashlight, and three skulls. The user is asked to pick up the right skull with his right hand, take the left skull with his left hand, and compare them to see if any differences are present. The narrator explains the differences between the two skulls: the morphological differences between a male and a female skull. In other words, sexual dimorphism. Finally, the user is invited to grab the skulls and place them on the osteometric table behind him or her so that measurements can be taken. The skulls uploaded in this virtual environment are those described above, the male and the female skulls with the magnified sexual characters.

In the second phase of the experience, the user is asked to play a mini-game, where a full human skeleton should be rebuilt starting from a group of commingled bones. Only the vertebral column is fixed, and the other pieces should be placed correctly. There is a reference skeleton near the operational field for simplifying the task. The competencies acquired in the first phase, his/her background knowledge, or even a little intuition should allow the users to easily reach the goal. However, the operator could interrupt the experience with a slightly different ending for those for whom the task seems too difficult. The experience correctly ends when the user places the bones in the correct locations, considering a generous margin of error (around 0.3 m from the correct place for each axis and 30 degrees for each rotation axis). This second phase of the experience highlights some aspects related to 'serious gaming', in particular, the following features are noticeable:

- Even if there is no time for completing the assigned task, the user is stimulated by the operator to give the best to complete the request as soon as possible;
- The correct displacement of a bone is highlighted with a green light and a sound;
- The final result is determined by the user: he/she can decide to rebuild the skeleton in the correct way or to play for fun; in both cases, personal satisfaction is the aim of the task. The only case in which the task is failed is when the user is still trying to place pieces in the correct locations, but the operator stops him/her, and the experience ends. The game mainly focuses on a non-punitive gameplay policy to allow anyone to enjoy the experience with an extremely low challenge level.

Some screenshots of the application can be seen in Figure 5.

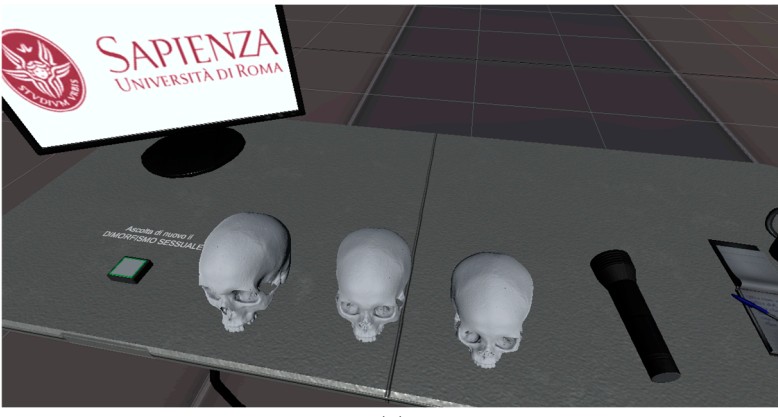

(**a**)

**Figure 5.** *Cont.*

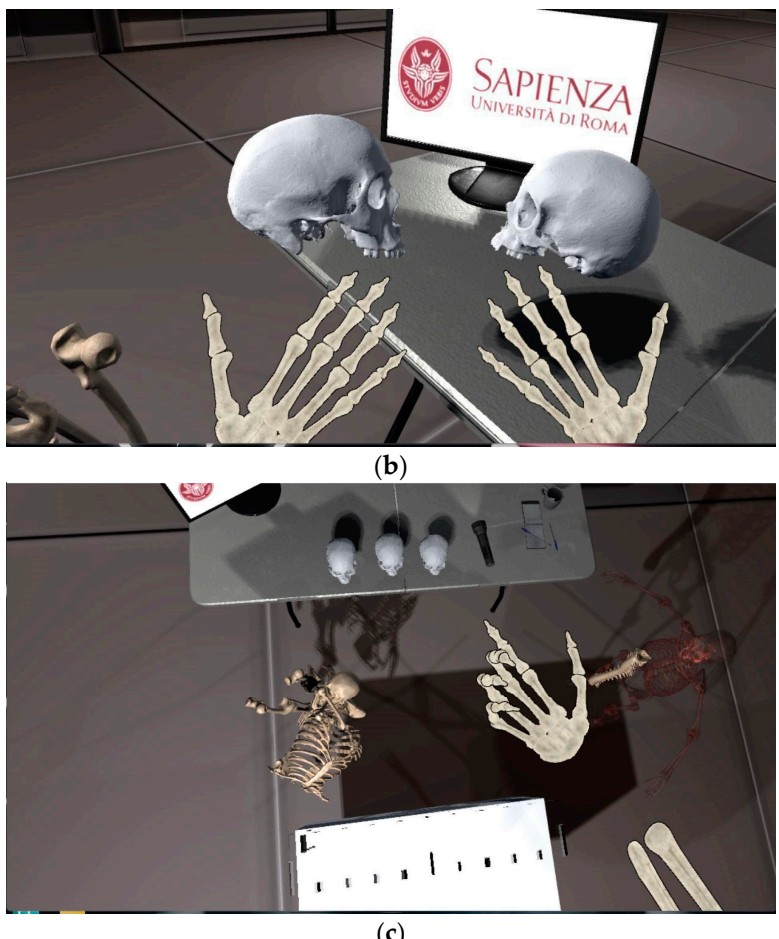

(**b**)

(**c**)

**Figure 5.** (**a**,**b**) Screenshots from the application's ego-view, while the user performs some actions in the virtual environment are shown. The last figure (**c**) is a top view of the operational space with a free-flight camera.

## 5. Evaluation Metrics and Results

This paper presents the results of empirical research based on user studies to evaluate the enjoyment and usability of a VR system dealing with biological anthropology and skeletal anatomy. The objective is to provide useful guidance for the most appropriate ways to virtually communicate and teach human osteology.

The first proposed assessment strategy focuses on evaluating the user's sense of presence experienced during the interaction with the VR device.

Whenever interaction with objects or the environment uses communication technology, the users are supposed to simultaneously perceive two distinct environments: the physical environment they are in and the digital environment presented through technology. To refer to the overall subjective experience with the virtual environment, the concept of "presence" is usually used. In VR environments, "presence" is usually defined as "the subjective experience of being in one place or environment, even when physically located in another." [36]. Presence is a subjective psychological response; it is individual and context-dependent; it depends on the user's mental imagery and ability to isolate themself from events occurring outside of the virtual environment. Presence is a subjective, individual, context-dependent psychological response [37]. Moreover, the Visitor Experience (VX) should stage experiences that satisfy visitors' desires to broaden the target audience [38]. To encourage museum visitors to repeat and recommend the visit, the experience must include a combination of experiential domains: education, entertainment, escapism, and aesthetics [39]. VR technologies can effectively create an experience that should simplify

learning, achieve new entertainment expectations, enhance the aesthetic experience, and contribute to escapism from reality.

There are also two other types of evaluation: emotional and cognitive. The emotional value (valence) of the experience will be reflected in a positive or negative cognitive evaluation by the visitor, who will also be more likely to repeat and recommend the experience. A comprehensive measure of Attitude Towards the Experience (ATE) is used to evaluate the valence of hedonic experiences that are chosen for pleasure and are affective and sensory in nature, as opposed to utilitarian experiences that are goal-oriented and cognitive in nature [40,41]. A VR experience falls into the first category.

Considering this context, we exploited the work of Leopardi et al. [42], where those parameters are involved in the experimental setup. We adopted the same questionnaire proposed by the authors and collected results according to their evaluation metrics.

Then, we added another parameter related to cybersickness: one of the most used metrics for this aim is the System Usability Scale (SUS) questionnaire. For shortness, in Table 2, only the SUS questions are shown. However, the other questionnaires can be found in the experimental section of [42].

**Table 2.** Questions in the SUS questionnaire.

| |
|---|
| I think that I would like to use this system frequently. |
| I found the system unnecessarily complex. |
| I thought the system was easy to use. |
| I think that I would need the support of a technical person to be able to use this system. |
| I found the various functions in this system were well integrated. |
| I thought there was too much inconsistency in this system. |
| I would imagine that most people would learn to use this system very quickly. |
| I found the system very cumbersome to use. |
| I felt very confident using the system. |
| I needed to learn a lot of things before I could get going with this system. |

The sample of respondents covers an age range of 27 to 62 in a collection of 60 participants. Regarding presence, the scale goes from 1 to 5, obtaining an average value of 4.18, which is a promising result compared with other similar systems in the state-of-the-art. Similarly, VX and ATE, on a scale between 1 and 7, scored an overall value of 5.42 and 6.10, respectively. These results seem to promote the system's goal of disseminating cultural content through an interactive and immersive experience.

Concerning the SUS (System Usability Scale), the overall score is approximately 70, corresponding to "Good", with a Standard Deviation of 6.25, highlighting a noticeable result considering that the system involves a VR-based tool. More details for each metric are reported in Table 3.

The results were obtained with devices produced in 2017, a discontinued Oculus CV1; thus, the scores could have been higher with cutting-edge technology.

However, we also collected some observations and comments that underlined some strengths and weaknesses of the proposal. About positive feedback, almost any user at the end of the experience was surprised at how interesting anthropological content could be with the support of enhancing technology combined with dedicated storytelling. This fact highlights the effectiveness of the proposed system for knowledge transfer when the focus contents could be considered "tedious" in advance. On the contrary, some users shared some comments about the grabbing action's lack of precision and accuracy. Even if the environment was fairly under control (low sunlight from the window, no other radio frequencies pointing at the LeapMotion device, smartphones in airplane mode), the hand tracking function sometimes fails to correctly locate the joints. This fact implies a

lack of smoothness in the user's actions while in play. Some possible solutions could be provided in future versions of this application: hand tracking sensors upgrade, more robust algorithms for hand tracking, and more precise classifiers for gesture recognition. Moreover, this experimental setup did not consider nor evaluate visual impairments separately, but it could be an improvement to focus on in the future.

**Table 3.** Statistics calculated on the collected results for each proposed evaluation metric.

|  | SUS | Presence | VX | ATE |
|---|---|---|---|---|
| **Population Standard Deviation (σ)** | 7.09 | 0.27 | 0.65 | 0.52 |
| **Population Variance (σ2)** | 50.28 | 0.07 | 0.42 | 0.28 |
| **Sample Standard Deviation (s)** | 7.15 | 0.27 | 0.65 | 0.53 |
| **Sample Variance (s2)** | 51.13 | 0.07 | 0.42 | 0.28 |
| **Coefficient of Variance** | 0.10 | 0.06 | 0.12 | 0.09 |
| **Mean (Average) (μ)** | 70.61 | 4.18 | 5.42 | 6.10 |

As a final thought, these considerations open a new horizon for future research and tests with users.

## 6. Conclusions

In this paper, a system for anthropological content dissemination is proposed. Modern technology allows the enjoyment of intense experiences in two directions: storytelling and natural interaction. In this context, the developed system exploits all the available functionalities according to the most recent standards in the state-of-the-art for promoting knowledge exchange in this cultural heritage field. The pipeline of the study is made of two main steps: the reconstruction of 3D digital models of skulls and the interactive environment development. The first phase involved the proposed MoldeR approach for exploiting machine learning-driven techniques to highlight sexual dimorphism and provide accurate meshes of the models. The second involved the development of an interactive environment in Unity3D, allowing a user to explore the contents in VR with a touchless approach. The obtained results highlight the project's effectiveness in content quality and usability. The users' questionnaires showed high scores in the presence, the significance of the exchanged knowledge, and the overall experience appreciation, promoting the system for this purpose. This fact implies that future research involving the latest hardware, more sensors, and improved procedures for 3D model reconstruction and natural interaction in virtual environments can be explored in the future. In general, we can affirm that the proposed work seems a promising idea that could be considered the ground base for a springboard in coming experimentations in this field, including novel technologies and algorithms.

**Author Contributions:** Conceptualization, R.M., S.G.M. and M.R.M.; methodology, R.M., S.G.M., D.A., L.C., A.D.B., L.L. and M.R.M.; software, D.A., A.D.B. and M.R.M.; validation S.G.M., D.A., L.C. and L.L; writing—original draft preparation, R.M., A.D.B. and M.R.M.; writing—review and editing, S.G.M., D.A., L.C. and L.L. All authors have read and agreed to the published version of the manuscript.

**Funding:** This work was supported by the "Smart unmannEdAeRial vehiCles for Human likE monitoRing (SEARCHER)" project of the Italian Ministry of Defence (CIG: Z84333EA0D).

**Acknowledgments:** The authors would like to thank Antonio Profico (Department of Biology— University of Pisa—Italy), Mary Anne Tafuri (Department of Environmental Biology—Sapienza University of Rome, Italy), and the Museum of Anthropology "G. Sergi" of Sapienza University of Rome, Italy.

**Conflicts of Interest:** The authors declare no conflict of interest.

## Notes

[1] AB = (x2 − x1)2 + (y2 − y1)2 + (z2 − z1)2
where $x$, $y$, and $z$ are the three spatial positions of points $A$ and $B$ in the virtual environment. If the value of all four distances is lower than a threshold (empirically set to 0.01 m), a grabbing action is performed, and each object that collides with the hand is attached to it.

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
