# Peer review of "Enhancement and Communication of Ancient Human Remains through VR: The Case Study of Sexual Dimorphism in the Human Skull"

_heritage, doi:10.3390/heritage6050217_

Round 1

Reviewer 1 Report

Thank you for submitting this interesting new research for review. My suggested edits and feedback are to be found in the attached marked-up version of your manuscript.

You have described what seems to be the beginnings of a potentially interesting use of VR within the sphere of osteology and public archaeology in a museum setting. However, your paper needs work to make your overall argument clearer. Is this a VR project or a 'serious game'? If a game, then what are the rules and mechanics beyond just articulating a skeleton or understanding general cranial sexual dimorphism?

I think the paper would be improved not only by using a table or two to more clearly present the initial results of user surveys, but also by including examples of the impressions or suggestions of test users, or by discussing where the project might go?

Compared to the examples you cite in the literature, this use of VR seems more suited to University classrooms or teaching hospitals. So, you should clarify the intended end-users and audience as well. 

In general, there is potential in this paper and project, but it doesn't feel quite finished yet. If this is merely the first step in something larger, then make sure to clearly indicate this. With these changes to content and the other edits, it will be in better shape.  

Although there are many areas (indicated on the marked up PDF provided) in which rewording or clarification would help you get your message across, in general, your use of English in this article is in general professional and of high quality. 

Author Response

Dear Reviewer,

thank you for your precious comments. In the attachment, you can find our answer to your comments.

Best regards,

Marco Raoul Marini

Reviewer 2 Report

The manuscript proposed a system for anthropological content dissemination. The main question addressed by this research was to exploit virtual reality to increase the experiences of visitors or researchers in interacting with virtual skulls in an interactive environment.

The article is worth publication if some concerns are addressed accordingly.

There are few comments here:

Page 3. Please use the full word before the abbreviation 3D

Page 5. R language (version? Stable release?)

Page 5. How do authors record the 3D coordinates? Using which application software?

Page 6. What is the results of 3D coordinates of 50 anatomical points (landmarks) and 200 geometrical points (semi-landmarks) taken on 163 samples?

Page 6. Why is magnification performed? Any relation with the final virtual skull?

Page 7. Figure 3. The 3D models with a factor of magnification of 1 and 2 showed similar size even after the reviewer magnified the image. In fact, the 3D models of the male and female also looked the same, looking at the remaining tooth and the size of orbit. The 3D model of the skulls are different than the one presented in Figure 5.

Page 8. Please use the full word before the abbreviation HMD. Only mentioned once in the abstract but not in the text. Similar to IR.

Page 8. HMDI cables? Do you mean HDMI? High-Definition Media Interface?

Page 9. The second phase, instead, involves a mini-game, where the user should rebuild… consider rewords this phrase

Page 10. Figure 5. There are 3 images altogether. Please label the image one by one and explain in the figure caption for easier understanding.

Page 11. Results should be presented in table or graph for easier understanding. Only average was given for the 60 participants. Any standard deviation?

Page 11. About Presence…. Why capital P?

Page 11. Please use dot instead of comma for the value e.g. 5,42 and 6,10 to 5.42 and 6.10.

Page 12. The second, instead, involved the development of an interactive environment in Unity3D, allowing a user….. instead?

Page 13 and 14. The format of references are not standardized.

Moderate editing of English language is needed

Author Response

Dear Reviewer,

Thank you for your precious comments. In the attachment, you can find our answer to your comments.

Best regards,

Marco Raoul Marini

Reviewer 3 Report

The paper entitled "Enhancement and communication of ancient human remains through VR: the case study of sexual dimorphism in the human skull" is well-constructed and interesting. The link between Virtual Reality and Anthropology is innovative and with great possibility of wide application. The technique using is repeatable and non-invasive.

The authors could improve the results explaining better the questions posed to respondents and inserting the proposed form.

The English is fluent and easy to read, the authors must correct same spelling mistake.

Author Response

(The authors gave the same response as above.)

Round 2

Reviewer 1 Report

This is a much improved manuscript, with only very few further small edits suggested. I appreciate the inclusion of additional contextual information, tables, and footnotes. 

The quality and overall use of English is much improved. No real issues noted. 

Author Response

Dear Reviewer,

Thank you again for the strong support provided in the review process. We modified our manuscript according to the second-round suggestions in the PDF, highlighting in green the changes (while keeping in red the changes of the first round).

Best regards,

Marco Raoul Marini